# Assessing biomarker trajectories for mortality risk in peritoneal dialysis: A focus on multivariate joint modeling

**Merve Basol Goksuluk[1], Dincer Goksuluk[1,2]\*, Murat Hayri Sipahioglu[3]**

**1** Department of Biostatistics, Faculty of Medicine, Sakarya University, Sakarya, Turkey, **2** Department of Biostatistics, Faculty of Medicine, Erciyes University, Kayseri, Turkey, **3** Department of Nephrology, Faculty of Medicine, Erciyes University, Kayseri, Turkey

\* dincer.goksuluk@gmail.com

**Data availability statement:** The peritoneal dialysis data supporting the findings of this

## Abstract

This study investigates mortality risk prediction in peritoneal dialysis (PD) patients through longitudinal biomarker analysis, comparing traditional and advanced statistical approaches. A retrospective cohort of 417 PD patients followed up between 1995 and 2016 at Erciyes University was analyzed, with serum albumin, creatinine, calcium, blood urea nitrogen (BUN), and phosphorus assessed as predictors of all-cause mortality. Statistical methods included Cox proportional hazards models, time-dependent covariates, and joint modeling (univariate and multivariate) for longitudinal-survival data integration. Joint models outperformed baseline, averaged, and time-dependent methods, with multivariate joint modeling yielding the highest predictive accuracy by incorporating inter-biomarker relationships. Serum albumin emerged as the most consistent mortality predictor, while creatinine and phosphorus showed significance in specific contexts. Other biomarkers, such as calcium and BUN, were less predictive. Dynamic prediction capabilities of joint models demonstrated enhanced alignment with patient outcomes, underscoring their utility in personalized medicine. This study highlights the importance of integrating temporal changes and biomarker interdependencies into survival analysis to improve risk stratification and clinical decision-making in PD patients. Future research should explore the broader applicability of these methods across diverse chronic disease populations.

## Introduction

End-stage renal disease (ESRD) marks the irreversible decline of kidney function, rendering the kidneys incapable of performing their essential roles in filtering waste products, excess fluids, and electrolytes from the body [1]. Consequently, these substances accumulate, leading to severe and potentially life-threatening health complications. ESRD represents the terminal phase of chronic kidney disease (CKD), a progressive condition characterized by the gradual deterioration of renal function. In advanced CKD, kidney function diminishes to a critical threshold, necessitating treatments such as dialysis or kidney transplantation for

study are available from the Department of Internal Medicine, Division of Nephrology, at Erciyes University (via https://tip.erciyes.edu.tr) upon reasonable request and subject to approval by the Institutional Executive Board, due to ethical and legal restrictions related to patient privacy and General Data Protection Regulation (GDPR) requirements. Researchers may also contact the senior author (Dr. Murat Hayri Sipahioglu, https://avesis.erciyes.edu.tr/mhsipahioglu), who can facilitate the data access process by forwarding requests to the appropriate institutional committee.

**Funding:** The author(s) received no specific funding for this work.

**Competing interests:** The authors have declared that no competing interests exist.

patient survival. While kidney transplantation offers superior survival rates and improved quality of life, its widespread implementation is constrained by factors such as patient eligibility, donor availability, cultural attitudes toward organ donation, local expertise, and the high costs of surgical procedures and immunosuppressive therapies. According to Mudiayi et al. [2], the global rate of kidney transplants is approximately 255 (58–432) per million population. In cases where transplantation is not viable, dialysis remains the primary alternative. The two main modalities of dialysis are (i) hemodialysis (HD) and (ii) peritoneal dialysis (PD). As of 2020, approximately 89% of dialysis patients worldwide undergo HD, while 11% receive PD [3]. The selection between PD and HD typically depends on factors such as the patient's medical requirements, lifestyle, proximity to an HD center, personal preferences, or physician recommendations. Sometimes, patients may attempt both methods to determine the most suitable option. Sinnakirouchenan and Holley [4] have suggested that PD may be associated with a lower mortality risk compared to HD during the initial stages of treatment; however, these risks appear to be similar over time.

The progression of PD patients is influenced by a range of prognostic factors, including age, diabetes, cardiovascular comorbidities, biomarkers such as serum albumin, C-reactive protein (CRP), estimated glomerular filtration rate (eGFR), etc. To monitor disease progression in dialysis patients, clinicians routinely collect a range of laboratory and clinical measurements at multiple time points during follow-up. The dynamic changes in these biomarkers over time –such as increases or decreases in their trajectories– play a critical role in guiding clinical decisions, including the need for additional interventions or modifications to treatment strategies. Statistical evaluations of these biomarker trajectories are used to assess whether the observed changes warrant clinical action. In such analyses, repeated measurements from PD patients are often incorporated to provide a comprehensive picture [5]. For example, serum albumin levels, a key indicator of nutritional and overall health status, undergo dynamic fluctuations during the PD process [6,7], as albumin levels can decline sharply and reversibly during acute illnesses [8].

Recent studies have employed both traditional and advanced prognostic models to predict mortality and adverse outcomes in PD. Traditional prognostic models, like Cox proportional hazards regression, often rely on static baseline or averaged biomarker values, which may fail to capture temporal variations critical for individualized risk prediction [8–13]. Approaches that average measurements or rely on single time points yield results that reflect population-level outcomes, potentially overlooking individual variability. Therefore, robust statistical models that integrate longitudinal data across all time points are essential for providing accurate and clinically relevant insights [14]. Recent advancements in joint modeling (JM) techniques, an advanced dynamic prognostic model that integrates longitudinal biomarker trajectories with survival data, have demonstrated superior predictive accuracy in PD patients, improving mortality risk prediction compared to conventional methods [15]. Similarly, Ma et al. [16] and Noh et al. [17] applied deep neural network algorithms to real-world PD datasets, demonstrating superior prediction accuracy and interpretability through feature importance recalibration.

Beyond nephrology, dynamic prediction models have demonstrated substantial utility in managing other chronic diseases by incorporating time-dependent patient data to refine prognostic accuracy. The paradigm of individualized risk prediction has gained prominence in modern medicine, particularly in the context of chronic disease management. Dynamic prediction models, which are updated as new information becomes available, enable personalized monitoring, screening, and intervention strategies over a follow-up period [18]. In oncology, for example, Wang et al. [19] developed a dynamic prediction model for patients with

advanced non-small-cell lung cancer that included evolving predictors such as Eastern Cooperative Oncology Group (ECOG) performance status and serum albumin. Their approach outperformed static models in forecasting short-term survival and supported individualized end-of-life care decisions. Similarly, Du et al. [20] constructed a landmark supermodel for esophageal cancer using data from the Surveillance, Epidemiology, and End Results (SEER) database, capturing time-varying covariate effects to dynamically update survival probabilities and guide treatment over follow-up. In type 2 diabetes, risk models using repeated biomarker measurements provided better calibration and discrimination than traditional models [21]. Collectively, these examples illustrate how trajectory-based dynamic models are increasingly being adopted across chronic disease domains to support personalized care and improve clinical outcomes.

This study aims to identify appropriate analytical methods for incorporating repeated measurements from PD patients over a specified follow-up period to assess their health status and inform clinical interventions. Additionally, it seeks to evaluate the performance of conventional techniques in the literature compared to advanced methods, emphasizing the importance of generating individualized outcomes.

## Materials and methods

### Study design and participants

This retrospective study was conducted at the Department of Nephrology of Erciyes University Hospital, TURKIYE, and utilized research data from a previously published paper [22]. At the time of data collection, obtaining ethical approval, particularly for retrospective studies, was not required by the Local Ethics Committee of Erciyes University. Consequently, the original study was conducted without ethical approval. Since we utilized data from an already published study and the data collection process was completed beforehand, obtaining ethical approval was not possible. Research data included 511 patients who initiated peritoneal dialysis (PD) at the Erciyes University Nephrology Department between March 1995 and June 2007 [22]. Data collection for some patients continued until November 2016. Patients were followed from the initiation of PD until the occurrence of one of the following events: death, kidney transplantation, discontinuation of PD, loss to follow-up, or the end of the study, whichever occurred first. Of the initial cohort, 423 patients met the inclusion criteria. Exclusion criteria were as follows: (i) recovery of kidney function, eliminating the need for further treatment; (ii) being under 18 years of age at the time of PD initiation; (iii) survival of fewer than 90 days after starting PD since this threshold is typically a minimum exposure period to establish a meaningful association between PD treatment and mortality outcomes [23,24]; (iv) pregnancy; (v) existence of other fatal diseases associated with short life expectancy (e.g., malignant tumor, severe cardiovascular disease, etc.); and (vi) missing data for the majority of key variables.

Additionally, to assess model performances, two test patients –one censored and one who died– were randomly selected without prior consideration of their biomarker trajectories (e.g., albumin, creatinine, calcium, phosphorus, and blood urea nitrogen), ensuring an unbiased evaluation. Selecting additional patients was unnecessary, as dynamic survival models generate patient-specific predictions by fitting at the individual level. Dynamic survival probabilities were estimated for these patients using each model evaluated in the study. A training dataset of 415 patients was used to develop and fit all survival models, providing a robust basis for comparing predictions.

## Clinical outcome

The primary outcome of interest was all-cause mortality. Patients who died while on PD or within three months after transitioning to HD were classified as PD-related deaths; all other cases were censored. The collected data included demographic information such as age at PD initiation, body mass index (BMI), gender, cause of ESRD, the presence of renal and comorbid conditions, hemodialysis history, and other relevant clinical factors. Clinical and biochemical measurements, including serum albumin, blood urea nitrogen (BUN), creatinine, calcium, phosphorus, peritoneal membrane transport characteristics, and peritonitis rates, were extracted from medical records. Additionally, a composite variable, "total count of comorbid and renal diseases or the number of illnesses (TCRD)," was defined as the sum of concurrent comorbid and renal conditions in each PD patient to assess their impact on mortality. TCRD, with possible values ranging from 0 to 9, encompasses diabetes mellitus, coronary artery disease, chronic heart failure, cerebrovascular disease, lung disease, liver disease (i.e., hepatitis), hypertension, polycystic kidney disease, and glomerulonephritis.

Time-varying biomarkers of interest included serum albumin, creatinine, calcium, BUN, and phosphorus, which were measured approximately every two months during the follow-up period. For statistical models, measurements recorded at six-month intervals were used. The study investigated the association between the trajectories of these longitudinal biomarkers and mortality, with a particular emphasis on their predictive value.

## Statistical analysis

All statistical analyses were performed using the R programming language (R Core Team, 2023), with the following packages: `survival` (version 3.8-3) [25,26], `JMbayes2` (version 0.5-0) [27], and `dynpred` (version 0.1.2) [28]. The normality of continuous variables was assessed using the Shapiro-Wilk test and graphical methods such as histograms, Q-Q plots, and box plots. Normally distributed variables were summarized using means and standard deviations, while nonnormally distributed variables were summarized using medians and minimum and maximum values. Categorical variables were reported as frequencies and percentages. Group comparisons for continuous variables were conducted using either the independent t-test or the Mann-Whitney U test based on the normality assumptions of the data. Categorical variables were compared using Chi-square tests, with Pearson's Chi-square or Fisher's Exact test applied as appropriate.

To investigate the association between longitudinal biomarker trajectories (i.e., serum albumin, blood urea nitrogen, creatinine, calcium, and phosphorus) and mortality in PD patients, we employed five distinct modeling approaches (M1 to M5), adjusting for potential confounders, including demographic, clinical, and biochemical measurements. Biomarkers were modeled as a univariate response in M1 to M4 and a multivariate response in M5. Initially, univariate Cox Proportional Hazard (PH) regression was applied to each potential confounder to identify risk factors significantly associated with mortality at a significance level of $p < 0.10$. These selected risk factors were incorporated as covariates in all subsequent survival models. The modeling approaches utilized are as follows:

- **M1 (Baseline)**: Cox PH models were constructed using baseline (initial) values of each biomarker, combined with the selected risk factors. Separate models were fitted for each biomarker.
- **M2 (Average)**: Time-averaged values of each biomarker, computed across the follow-up period, were included in Cox PH models alongside the selected risk factors, with separate models for each biomarker.

- **M3 (Time-Dependent)**: Time-dependent Cox PH models incorporated the longitudinal values of each biomarker as time-varying covariates, adjusted for the selected risk factors. Separate models were fitted for each biomarker.
- **M4 (Univariate Joint Modeling)**: For each biomarker, a univariate joint model integrated a mixed-effects model for the longitudinal trajectory with a Cox PH survival model, adjusted for the selected risk factors. Separate models were fitted for each biomarker.
- **M5 (Multivariate Joint Modeling)**: A multivariate joint model simultaneously analyzed all biomarkers, using a multivariate mixed-effects model for their longitudinal trajectories and a Cox PH survival model, adjusted for the selected risk factors. A common model were fitted for all biomarkers.

We evaluated model performance using the area under the receiver operating characteristic (ROC) curve (AUC) for dynamic prediction. AUC values for baseline (M1), time-averaged (M2), and time-dependent (M3) methods were computed using the `dynpred` package, while the joint modeling approaches utilized the `JMbayes2` package in R. Significance level was set at 0.05 for all analyses. For mathematical background and further details of fitted models, M1 to M5, see Appendix A.

## Results

During the follow-up period, 87 patients (20.9%) died, while 330 patients (79.1%) were censored from further analysis. The median follow-up duration was 30 months, ranging from 3 to 137 months. Detailed demographic and clinical characteristics of the entire cohort and its respective subgroups are presented in Table 1. At the initiation of PD, the mean age was $45.96 \pm 14.33$ years, and the BMI was $23.63 \pm 4.11$ kg/m$^2$. Deceased patients were significantly older than censored patients ($p < 0.001$). A total of 50 patients (12.0%) had previously undergone HD before starting PD, with a significantly higher proportion in the deceased group ($p < 0.001$). Additionally, significant differences were observed between the deceased and censored groups regarding the total number of renal and comorbid diseases ($p < 0.001$), peritonitis rate ($p = 0.002$), serum albumin levels at PD initiation ($p = 0.048$), time-averaged serum creatinine ($p = 0.007$), and time-averaged serum calcium levels ($p = 0.024$).

Fig 1 illustrates the biomarker levels of patients with PD over the follow-up period. Each point represents a patient's biomarker value at specific time points. The red line indicates the average trend across all patients in the study ($n = 417$) over time, while the green lines show the values of two patients from the test set—one who died and one who was censored—across the same period. As shown in Fig 1, the average trends of the patients (red line) do not align with the individual trends of the two patients (green lines).

In the time-to-event analysis, univariate Cox PH regression identified several statistically significant predictors: age at PD initiation ($p < 0.001$), BMI ($p < 0.001$), history of prior HD ($p < 0.001$), total count of comorbid and renal diseases (TCRD) ($p < 0.001$), and peritonitis rate ($p < 0.001$). These predictors were included as confounders in all Cox PH submodels. Longitudinal biomarkers were incorporated into the Cox PH models using various approaches, as detailed in Table 2. Due to the extensive nature of the model results, only parameter estimates for the longitudinal biomarkers are reported in Table 2. Comprehensive results for all fitted models are presented in Appendix B. The proportional hazards assumption was assessed for all survival models using the `cox.zph` function in R. Global tests indicated no violation of the proportional hazards assumption; although the peritonitis rate showed local violations in some models, its inclusion was retained due to its clinical importance and the absence of global model invalidation. Given the potential inflation of Type I

**Table 1. Biochemical, clinical and demographic findings of study group.**

| Characteristic | Censored (n = 330) | Dead (n = 87) | Total (n = 417) | p value* |
|---|---|---|---|---|
| Age, *year* | 44.78 ± 14.30 | 50.32 ± 13.70 | 45.96 ± 14.33 | <0.001 |
| BMI, $kg/m^2$ | 23.5 ± 4.08 | 24.18 ± 4.18 | 23.63 ± 4.11 | 0.214 |
| Gender, *Female* | 142 (43.0) | 36 (41.4) | 179 (42.9) | |
| Cause of ESRD[†] | | | | |
| Diabetes mellitus (DM) | 106 (32.0) | 39 (45.4) | 145 (34.8) | 0.018 |
| Glomerulonephritis | 32 (9.7) | 6 (7.0) | 38 (9.1) | 0.452 |
| Hypertension | 53 (16.0) | 9 (10.5) | 62 (14.9) | 0.207 |
| Polycystic kidney disease (PKD) | 13 (3.9) | 6 (7.0) | 19 (4.6) | 0.219 |
| Unknown | 96 (29.0) | 16 (18.6) | 112 (27.1) | 0.057 |
| Other | 32 (9.7) | 9 (10.5) | 41 (9.8) | 0.806 |
| Comorbidity[†] | | | | |
| Cardiovascular disease | 57 (17.2) | 35 (40.7) | 92 (22.1) | <0.001 |
| Lung disease | 7 (2.1) | 6 (7.0) | 13 (3.1) | 0.019 |
| Hepatitis | 33 (9.9) | 27 (31.4) | 60 (14.4) | <0.001 |
| Transferred from HD, *Yes* | 30 (9.1) | 20 (23.5) | 50 (12.0) | <0.001 |
| Transportation characteristic, *High* | 162 (49.0) | 45 (52.9) | 207 (49.9) | 0.527 |
| Number of illnesses (TCRD) | 1 [0, 4] | 2 [0, 5] | 1 [0, 5] | <0.001 |
| Peritonitis rate, *episodes/patient-year* | 0.25 [0, 5.33] | 0.57 [0, 3] | 0.32 [0, 5.33] | 0.002 |
| Serum albumin, *g/dL* | | | | |
| Baseline | 3.60 ± 0.61 | 3.45 ± 0.55 | 3.58 ± 0.59 | 0.048 |
| Time-averaged | 3.55 ± 0.49 | 3.46 ± 0.36 | 3.53 ± 0.47 | 0.121 |
| Blood urea nitrogen, *mg/dL* | | | | |
| Baseline | 54.13 ± 19.16 | 54.17 ± 22.07 | 54.14 ± 19.77 | 0.985 |
| Time-averaged | 55.40 ± 13.30 | 51.97 ± 15.96 | 54.68 ± 13.94 | 0.426 |
| Serum creatinine, *mg/dL* | | | | |
| Baseline | 6.88 ± 3.11 | 6.75 ± 3.21 | 6.85 ± 3.12 | 0.728 |
| Time-averaged | 8.10 ± 3.03 | 7.13 ± 2.66 | 7.90 ± 2.98 | 0.007 |
| Calcium, *mg/dL* | | | | |
| Baseline | 8.94 ± 1.06 | 9.10 ± 0.98 | 8.98 ± 1.04 | 0.213 |
| Time-averaged | 9.08 ± 0.71 | 9.27 ± 0.66 | 9.12 ± 0.71 | 0.024 |
| Phosphorus, *mg/dL* | | | | |
| Baseline | 4.56 ± 1.48 | 4.76 ± 1.80 | 4.60 ± 1.54 | 0.277 |
| Time-averaged | 4.66 ± 1.23 | 4.67 ± 1.34 | 4.66 ± 1.25 | 0.935 |

Summarized using mean ± standard deviation, frequency (percentage) or median [minimum, maximum] where appropriate.

**Note**: Sample sizes were given in each group. Valid sample sizes might be different in each variable depending on the missing data.

* Chi-squared tests (i.e., Pearson's chi-squared or Fisher's exact), Student's t-test or Mann-Whitney U test were used depending on the parametric test assumptions.

**BMI**: Body mass index, $kg/m^2$; **ESRD**: End-stage renal disease; **HD**: Hemodialysis.

[†] Patients might have more than one disease causing ESRD.

error from multiple testing, modeling decisions were based on overall test patterns rather than isolated deviations. Among the biomarkers, serum albumin consistently demonstrated a significant association with all-cause mortality across all models, with a 1 mg/dL decrease in serum albumin leading to a 3.29-fold increase in the risk of death (95% CI: [1.63, 7.23])

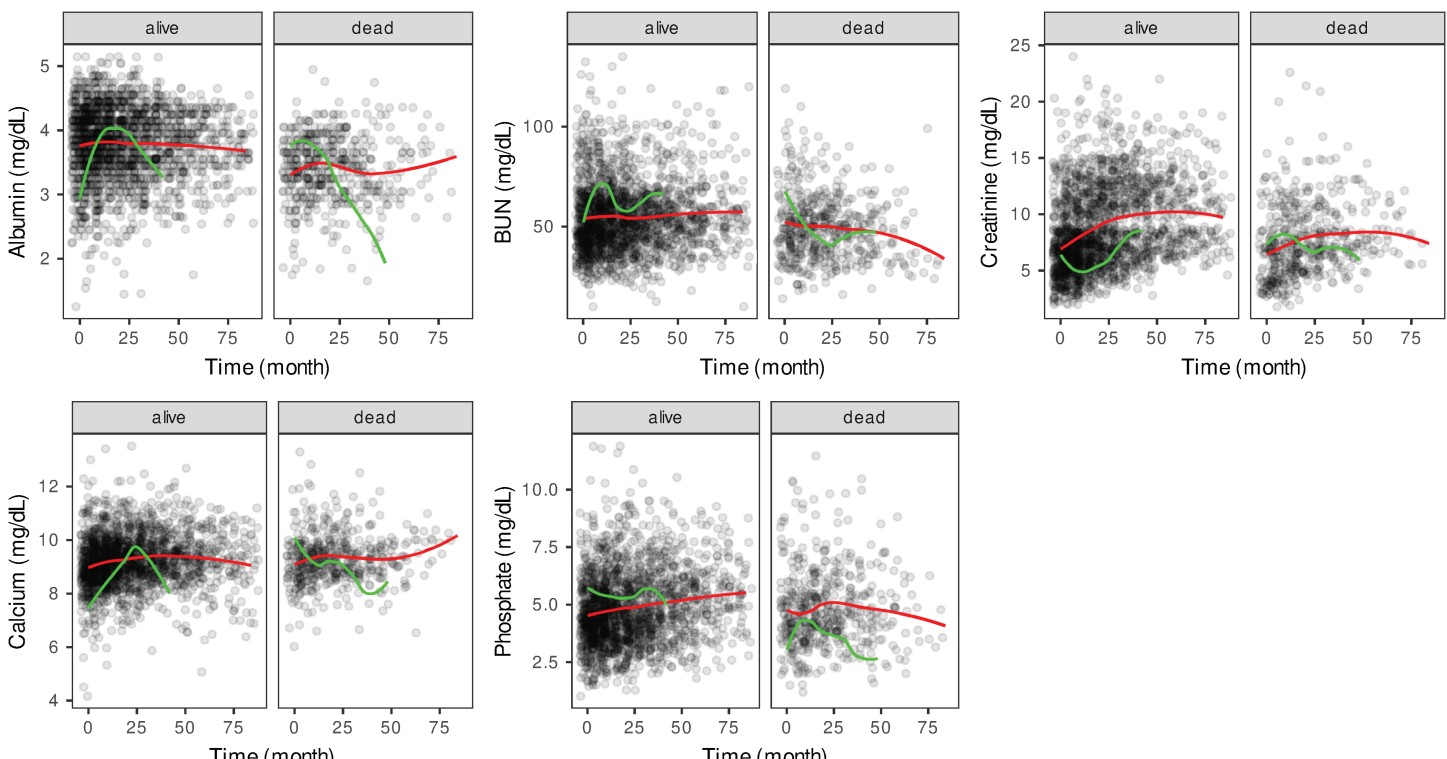

**Fig 1. Overall (red line) and patient-based (green line) changes of serum albumin, BUN, creatinine, calcium and phosphorus levels in follow-up period according to event status.**

in the multivariate joint modeling (M4). In contrast, BUN and calcium levels did not significantly predict mortality in PD patients ($p > 0.05$ in all models). Average serum creatinine levels were significant for mortality risk while the time-dependent Cox PH model (M3) and univariate joint modeling (M4) indicated that decreases in serum creatinine and phosphorus were associated with increased mortality risk. Notably, when considering the relationships among biomarkers, only serum albumin levels consistently remained a significant predictor of mortality in the multivariate joint modeling (M5).

Two patients, i.e., one deceased and one censored, were removed from the complete dataset to serve as independent test samples. We dynamically estimated the survival probability of these patients at future time points using each model and assessed the influence of longitudinal biomarker changes over the follow-up period. Fig 2 illustrates the dynamic predictions for the deceased (left column) and censored (right column) patients across each biomarker. The vertical dashed line represents the last time patient measurements were recorded, while the horizontal red line indicates the average value of repeated measurements during the follow-up period. Biomarker trajectories are shown to the left of this line, with predicted survival probabilities displayed to the right. As seen in Fig 2, the joint modeling approaches provided survival probability estimates aligned more closely with the patient's actual status than other methods. Both the U-JM and M-JM approaches predicted lower survival probabilities for the deceased patients and higher survival probabilities for the censored patient. Although U-JM and M-JM yielded similar results, U-JM predictions for deceased patients were more sensitive to changes in the biomarker levels over time.

**Table 2. The results of survival models with different forms of time-varying biomarkers (n = 415).**

| | Baseline (M1) | | Average (M2) | | TD-Cox (M3) | | U-JM (M4) | | M-JM (M5) | |
|---|---|---|---|---|---|---|---|---|---|---|
| | HR | | HR | | HR | | HR | | HR | |
| | (95% CI) | p | (95% CI) | p | (95% CI) | p | (95% CI) | p | (95% CI) | p |
| ALB | 1.54† (1.09, 2.17) | 0.01 | 2.01† (1.12, 3.60) | 0.02 | 2.16† (1.49, 3.13) | <0.001 | 2.90† (1.66, 5.13) | <0.001 | 3.29† (1.63, 7.23) | <0.001 |
| BUN | 0.99 (0.98, 1.01) | 0.94 | 0.99 (0.96, 1.01) | 0.20 | 0.99 (0.98, 1.01) | 0.14 | 1.02 (0.99, 1.04) | 0.12 | 1.01 (0.98, 1.04) | 0.46 |
| CRE | 0.94 (0.85, 1.04) | 0.21 | 1.26† (1.26, 1.41) | <0.001 | 1.16† (1.06, 1.27) | <0.001 | 1.16† (1.04, 1.29) | 0.001 | 1.1 (0.96, 1.25) | 0.15 |
| CA | 0.88 (0.71, 1.1) | 0.26 | 0.89 (0.62, 1.29) | 0.53 | 1.06 (0.83, 1.33) | 0.64 | 0.91 (0.73, 1.66) | 0.65 | 0.76 (0.45, 1.25) | 0.29 |
| P | 0.98 (0.86, 1.13) | 0.85 | 0.85 (0.71, 1.02) | 0.08 | 1.20† (1.04, 1.39) | 0.01 | 1.25† (1.02, 1.54) | 0.03 | 1.02 (0.77, 1.32) | 0.85 |

†Risk estimates were given for 1-unit decrease in measured variable if found statistically significant.
**ALB**: Serum albumin, *g/dL*, **BUN**: Blood urea nitrogen, *mg/dL*, **CRE**: Serum creatinine, *mg/dL*.
**CA**: Calcium, *mg/dL*, **P**: Phosphorus, *mg/dL*.
**HR**: Hazard Ratio, **CI**: Confidence Interval.

Fig 3 presents the dynamic AUCs, reflecting the methods' performance over 5-years from PD initiation. AUC values were calculated based on the probability of survival after 6 months for a patient known to be alive at a given time. The time-dependent Cox PH model performed comparably or slightly better than the baseline and time-averaged models at some time points. In contrast, the joint modeling approaches demonstrated superior performances overall, with M-JM performing slightly better than U-JM, particularly at specific time points. The highest model performance was observed at the 48-th month using joint modeling approaches, whereas the AUC for baseline (M1), time-averaged (M2), and time-dependent (M3) models fell below 0.5 by the 54-th time point. Performance fluctuations across methods could be attributed to variations in the patient's nutritional status and kidney functions during the follow-up period (Fig 2).

## Discussion

Repeated measurements in chronic diseases are essential for monitoring treatment effectiveness and disease progression, enabling timely intervention. While traditional statistical analyses often simplify repeated measurements to baseline or average values, such approaches may yield inconsistent findings. Incorporating longitudinal data directly into time-to-event models, such as time-dependent Cox regression, improves reliability but assumes error-free measurements, which may not reflect real-world data. Joint modeling has gained popularity as it simultaneously analyzes longitudinal and survival data, linking repeated measurements with survival outcomes.

This study contributes to the growing body of evidence supporting the use of joint models and trajectory-based predictors in PD. The observed superiority of multivariate joint models in our analysis aligns with prior work [16,29], which demonstrated that incorporating the temporal evolution and interrelationships of biomarkers leads to more robust and clinically meaningful predictions. In our study, we implemented subject-specific dynamic prediction models for mortality. In both univariate and multivariate forms, we demonstrated that joint

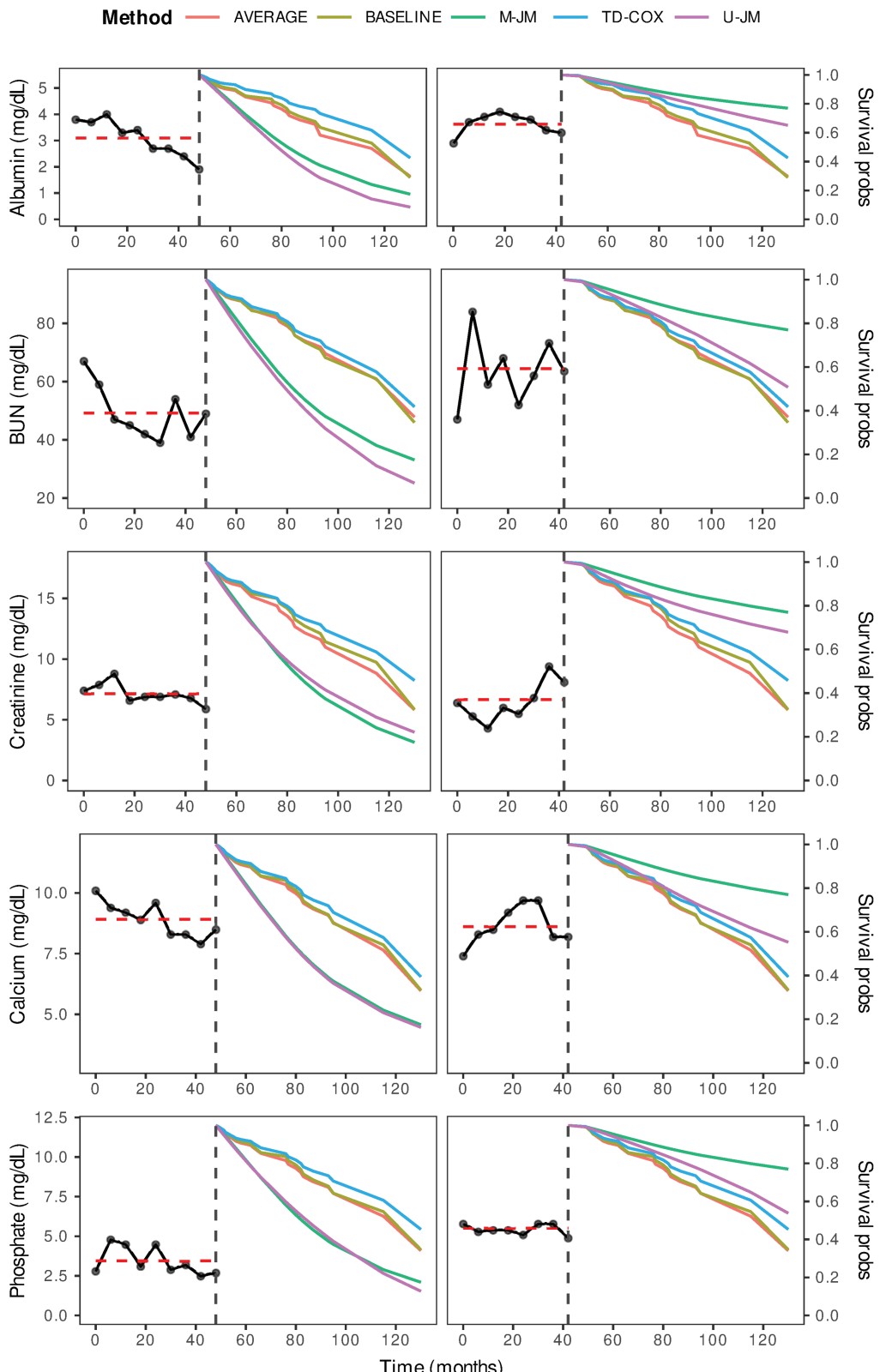

**Fig 2. The prediction of survival probability of two patients – left column: dead patient (age: 48, gender: male) and right column: censored patient (age: 32, gender: male).**

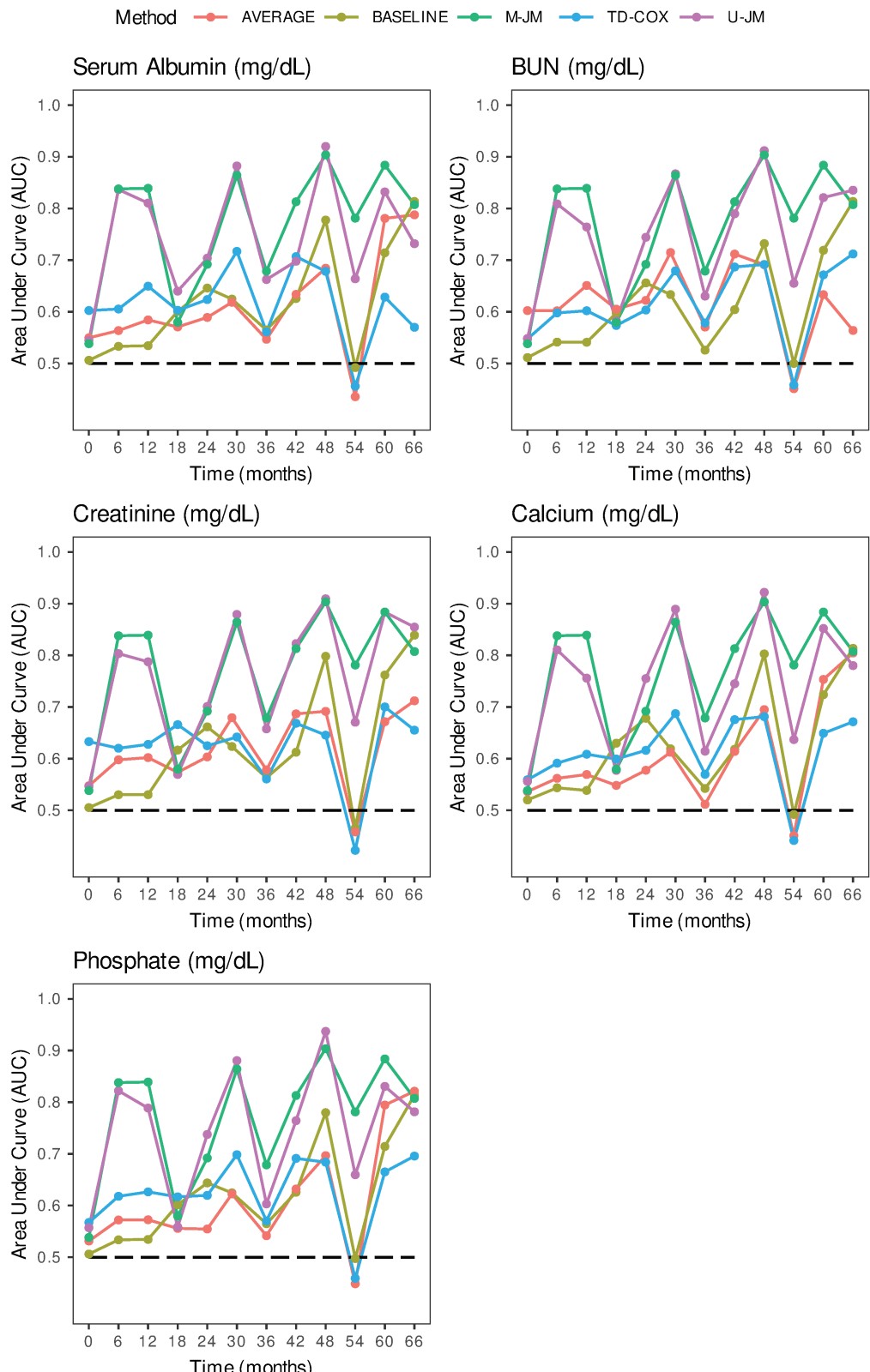

**Fig 3. The prediction of dynamic AUC levels in each model for each biomarker at six-month intervals.**

models outperformed other approaches, including baseline values, follow-up averages, and the time-dependent Cox model. Notably, the multivariate joint model generally provided superior predictive performance by effectively capturing the association between biomarkers. Additionally, our model incorporated instant changes observed during follow-up into mortality predictions, enhancing its accuracy and clinical utility. These findings underscore the value of using advanced joint modeling approaches for analyzing temporal biomarker changes and improving survival predictions in patients with ESRD undergoing PD.

Serum albumin levels reflect a range of factors, including protein and calorie intake, dialysis adequacy, peritoneal and renal albumin loss, concurrent illness, and underlying systemic disease [5]. Known as a reliable predictor of mortality risk in patients undergoing peritoneal dialysis (PD), serum albumin is used to assess nutritional status and indicate inflammation. Continuous monitoring of serum albumin levels is essential for improving survival in PD patients [6]. Consequently, studies on PD patients often incorporate serum albumin levels directly as a covariate in the model [30]. A study examining baseline serum albumin levels has shown that low levels are associated with an increased mortality risk [12]. Studies have also demonstrated that the time-averaged serum albumin levels measured at three-month [31], six-month [8,32], 12-month [33], and 24-month [30] intervals during follow-up significantly and negatively affect mortality risk. However, Kang et al. [10] did not find a significant association between mortality and the time-averaged serum albumin levels measured every six months over a five-year follow-up period (HR=1.033; p=0.916). Another study examining the effects of baseline and one-year average serum albumin levels found that while baseline serum albumin levels at the start of PD did not affect mortality risk, the one-year average levels significantly impacted mortality [11]. Çankaya et al. [13] further noted that baseline serum albumin alone may not successfully predict mortality; even if initial levels are low, increasing albumin levels during PD may reduce mortality. They also reported that time-averaged albumin levels provided more meaningful mortality predictions than baseline. Similarly, Song et al. [7] found that time-averaged albumin levels were superior to baseline levels in predicting mortality for PD patients. The dynamic changes in albumin over time held greater clinical significance for mortality risk than baseline albumin levels. Chiu et al. [5] investigated serum albumin trajectories by assessing albumin levels at the start of PD, peak albumin levels during PD, and end-of-PD levels. They concluded that albumin levels at these time points significantly impacted mortality and that analyzing serum albumin trends could be beneficial. One study that accounted for baseline and time-averaged albumin values found a relationship between baseline serum albumin levels and mortality. However, time-averaged values offered better predictive accuracy and prognostic insight [34]. Another study compared initial serum albumin values with the trajectory of serum albumin levels measured roughly every three months over three years fitted with U-JM results based on serum albumin levels measured approximately every three months over three years, concluding that the trajectory of serum albumin was the best model [14]. In studies that analyzed serum albumin levels using both time-dependent and U-JM approaches, changes in serum albumin over time were found to affect mortality risk in both the time-dependent model significantly (Coefficient: -0.58; $p < 0.001$) and the univariate joint modeling (Coefficient: -1.24; $p = 0.011$) [35]. Another study utilizing a U-JM reported that increases in albumin trajectories post-dialysis at any time point were associated with a reduced risk of death (HR = 0.881; $p < 0.001$), suggesting that albumin trajectories post-dialysis were better predictors of mortality than baseline serum albumin [36]. In our study, consistent with the literature, serum albumin level was identified as a significant predictor of mortality risk in all forms analyzed.

Calcium and phosphorus levels are crucial for bone health and mineral balance. When kidney function is impaired, regulating these two minerals becomes challenging. Mineral

metabolism disorders, such as abnormal serum calcium and phosphorus levels, are common in patients with chronic kidney disease [37]. Therefore, monitoring these values is essential for assessing kidney function and guiding treatment.

Phosphorus metabolism disorders are a common complication in end-stage renal disease (ESRD) and play a significant role in bone integrity, ectopic calcium deposition, calciphylaxis, vascular calcification, and myocardial hypertrophy. Approximately 40% of dialysis patients experience hyperphosphatemia [38]. Therefore, closely monitoring phosphorus levels is essential in patients receiving renal failure treatment. In a study involving PD patients, phosphorus levels were measured regularly over 120 months, with each time point analyzed separately. While baseline serum phosphorus levels were not associated with mortality risk (HR = 0.764; 95% CI: [0.521, 1.119]; $p$ = 0.166), phosphorus levels measured at the third-month post-dialysis significantly impacted mortality risk (HR = 1.666; 95% CI: [1.007, 2.758]; $p$ = 0.047) [33]. In Liu et al.'s study [39], high baseline serum phosphorus level was found to influence early mortality risk (<3 months on PD) (HR = 1.391; $p$ < 0.001). Additionally, the time-averaged phosphorus measurements taken every six months showed a significant association with mortality risk (HR = 2.64; 95% CI: [1.061, 6.566]; $p$ = 0.037) [10]. Rivara et al. [40] reported that higher time-averaged phosphorus levels (above 6.4 mg/dL) were linked to a significantly higher mortality rate. However, they did not observe an increased mortality risk at low serum phosphorus levels. Conversely, Yaghoubi et al. [41] found that patients with serum phosphorus levels below 4 mg/dL had a 1.6 times higher mortality risk than those with levels between 4–6 mg/dL (95% CI: [1.1 – 2.2]). Güler et al. [29] investigated the relationship between longitudinally collected calcium, PTH, creatinine, and phosphorus levels and mortality risk in PD patients using the M-JM approach to account for correlations among these biomarkers. While serum phosphorus level was not significant with the U-JM approach, it was a significant predictor of mortality risk when analyzed with the M-JM approach. In our study, phosphorus level was associated with mortality risk according to the time-dependent Cox and U-JM results, considering the trajectory over the follow-up period (HR = 1.20; 95% CI: [1.04,1.39]; $p$ = 0.013 and HR = 1.25; 95% CI: [1.02,1.55]; $p$ = 0.026, respectively), though it was not found to be significant with the M-JM approach.

Low serum calcium levels, which are crucial for bone health, can lead to muscle cramps and osteoporosis in patients with chronic kidney disease (CKD). Consequently, upon initiation of dialysis, patients are typically given vitamin D supplementation, calcium-based phosphate binders, and calcium in the dialysate solution to support calcium levels. Following dialysis initiation, an increase in serum calcium levels is expected. However, elevated calcium levels can lead to complications such as vascular calcification, underscoring the importance of closely monitoring these values throughout the follow-up period. In a study by Melamed et al. [42] investigating whether longitudinal changes in bone mineral parameters were independent of mortality, serum calcium levels were found to increase from baseline up to the sixth month, after which they stabilized. Calcium values were included in the analysis both as baseline measurements and in a time-dependent manner. While baseline calcium levels were not significantly associated with mortality risk, higher calcium levels in the time-dependent analysis affected mortality risk significantly. The time-dependent model revealed that calcium levels >9.73 mg/dL were associated with a 52% increase in mortality risk. Rivara et al. [40] examined serum calcium values averaged over 20 calendar quarters and reported that both low and high serum calcium levels (especially >10.2 mg/dL) were associated with increased mortality risk. In another study, which averaged the first three calcium measurements, no association was found between calcium levels and mortality risk [41]. Using the U-JM and M-JM approaches, Güler et al. [29] found no significant relationship between calcium trajectory and mortality risk. Similarly, Menon et al. [43] examined the relationship between

baseline phosphorus and calcium-phosphorus product levels and mortality risk in CKD stage 3–4 patients, concluding that neither serum phosphorus nor calcium-phosphorus product levels were associated with mortality rates. In our study, calcium levels were not significant predictors of mortality risk in any model.

Creatinine and BUN levels are frequently used in blood tests to evaluate kidney function, providing insights into how well the kidneys function and assess protein metabolism in the body. Mejia et al. [44] indicated that baseline albumin and creatinine levels were significant predictors of mortality risk in PD patients, suggesting that the nutritional status of patients at the start of dialysis is a critical factor in determining long-term outcomes. In a study investigating factors influencing mortality risk among PD patients, baseline and 12-month follow-up BUN and creatinine values were compared between deceased and surviving groups, finding no significant differences between groups at either time point ($p > 0.05$). However, the study noted that higher serum creatinine levels in PD patients have been associated with a higher likelihood of survival in some reports [45]. A study evaluating risk factors and clinical outcomes related to mortality in PD patients referenced the consideration of clinical significance, based on a literature review, to determine mortality predictors. Although BUN and creatinine values were collected, the model did not include these variables as mortality predictors [46]. In a 12-year study from Mexico on continuous ambulatory peritoneal dialysis (CAPD) mortality, initial albumin, and creatinine levels were significant in a univariate Cox PH survival model but not in a multiple Cox PH survival model. The authors stated that variables were selected based on clinical relevance from the literature. While BUN and calcium values were measured, they were not considered influential factors in mortality prediction [47]. In a Turkish study, baseline levels of albumin, BUN, creatinine, calcium, and phosphorus were measured; however, aside from albumin, none of these biomarkers significantly differed between deceased and surviving groups ($p > 0.05$) [48]. Block et al. [49] found that calcium levels, when adjusted for albumin, were not associated with mortality risk. Guler et al. [29] suggested that as creatinine reflects muscle mass and nutritional status, low creatinine levels often indicate protein loss and can strongly impact survival probability. In Guler et al.'s study, creatinine levels were not a significant predictor of mortality risk according to U-JM results, yet they were significant under the M-JM model. In our study, BUN values were insignificant predictors in any of the five models ($p > 0.05$). Creatinine levels, however, were significant predictors of mortality in both time-averaged and trajectory forms using the time-dependent Cox PH model and U-JM. Nevertheless, when the M-JM model accounted for relationships among other biomarkers, creatinine levels were insignificant in predicting mortality risk.

In our study, two patients were selected as a test set and excluded from the primary analysis. Using all models developed, we estimated the survival probabilities for these two patients at future time points beyond the final time point observed in the study. The results indicate that joint modeling approaches (U-JM and M-JM) provided probability estimates closer to the patients' actual outcomes. This finding is likely due to the joint modeling approach's statistically robust framework, which integrates longitudinal and survival data into a single model rather than handling these data separately. Another advantage of the joint modeling approach is its capacity for dynamic predictions by updating risk estimates as new longitudinal data become available. This feature allows the recalculation of risk estimates at each time point based on the latest patient data, supporting personalized medicine by enabling tailored risk predictions. Although U-JM and M-JM demonstrate similar performance, considering the biomarker interrelationships could influence results. By incorporating correlations and dependencies between biomarkers during modeling, the M-JM method enables more accurate

predictions, mainly when strong correlations exist between biomarkers [29]. BUN and creatinine levels are essential indicators for assessing kidney function, while calcium and phosphorus levels are key indicators of bone metabolism and overall mineral balance, highlighting the importance of accounting for the relationships among these markers. Huang et al. [38] reported that serum albumin levels influence the association between phosphorus levels and mortality; specifically, higher albumin levels mitigate the mortality risk associated with elevated phosphorus. Therefore, they emphasized the need to simultaneously evaluate the interactive effect of serum albumin and phosphorus for clinical outcomes, particularly in newly dialyzed patients. Additionally, Rivara et al. [40] noted that nearly half of circulating calcium is albumin-bound, and those with the lowest calcium levels often also have the lowest serum albumin levels, indicating that the association between calcium and mortality risk is substantially affected by albumin levels.

The clinical relevance of methods we have utilized in this study is evident in their ability to provide patient-specific, time-updated survival probabilities, supporting proactive care strategies. For example, Ma et al. [16] illustrated how adaptive dynamic prediction models identified PD patients at rising mortality risk, allowing for targeted intervention based on changing biomarker profiles. While this study focused on PD, the value of dynamic prediction is also evident in other chronic conditions. Naumzik et al. [50] demonstrated the utility of dynamic treatment planning in chronic diseases by distinguishing stable and unstable disease phases using trajectory modeling. Similarly, in chronic heart failure and oncology, joint modeling frameworks have facilitated early risk identification and informed timely clinical responses, reinforcing the generalizability of these approaches.

This study has several limitations. Notably, we were unable to incorporate the Charlson Comorbidity Index (CCI) into our models due to missing data for key variables required to compute CCI scores for most subjects. Instead, we used the "total number of comorbid and renal diseases (TCRD)" as a proxy to account for comorbidity burden. While TCRD captures some aspects of renal and comorbid conditions, it does not fully reflect the comprehensive impact of CCI, potentially limiting the precision of our comorbidity adjustments. Second, the absence of a simulation study limits our findings' generalizability. Conducting such a study was impractical due to differing model distributional assumptions, risking biased data favoring specific models, with overparameterization and computational issues. Under favorable conditions, a comprehensive simulation study can be conducted, for instance, by varying sample sizes, predictors, multivariate responses, and baseline hazard specifications, to enhance applicability. We leave this as a further research topic.

## Conclusion

The changes observed in repeated measurements collected during the follow-up period serve as critical indicators for predicting mortality risk in patients undergoing peritoneal dialysis. Accordingly, utilizing comprehensive data from repeated measurements, such as the trajectory of biomarkers, rather than relying on a single measurement, such as baseline or average values, enhances the predictive performance of the model and, consequently, the reliability of the results. Given the increasing significance of personalized medicine, the joint modeling approach, which enables the simultaneous modeling of longitudinal and survival data for patient-specific dynamic predictions, is recommended. When analyzing multiple time-varying biomarkers simultaneously, the interrelationships among these biomarkers should also be incorporated into the analysis in the context of multivariate response models.

# Appendix

## A. Mathematical background of fitted survival models - A brief explanation

The peritoneal dialysis (PD) data were analyzed using the modeling approaches (M1 – M5) described in the Statistical Analysis section. To ensure fair and consistent comparisons across models, variable selection followed a standardized procedure: (i) risk factors influencing survival were uniformly included across all models, identified via univariate Cox proportional hazards (PH) regression ($p < 0.10$). In joint modeling approaches (M4 and M5), time-varying biomarker effects were adjusted for potential confounders before inclusion in the survival submodel. While the survival submodel retained consistent risk factors, the longitudinal submodels in M4 and M5, using linear mixed-effects models, allowed flexible selection of confounders specific to each biomarker's time-varying trajectory, enhancing model adaptability. The models were categorized based on the response type (univariate or multivariate) and the nature of biomarker effects (time-varying or time-independent). The general specifications of the models (M1 – M5) are summarized in Table 3.

Potential confounders and longitudinal biomarkers influencing survival were examined through the models detailed in Table 3. The following subsections briefly provide technical and methodological details for each model.

**Cox PH Models (Baseline, Averaged, and Time-dependent; i.e., M1, M2, and M3).** The effects of longitudinally measured biomarkers ($X_b$) on survival were initially examined using Cox PH models, as defined in Eq (1), without accounting for time-varying effects.

$$h(t \mid \mathbf{X}) = h_0(t) \cdot \exp\left(\mathbf{X}_c \beta_c + X_b \beta_b\right) \tag{1}$$

Here, $h_0(t)$ represents the baseline hazard function, and $\mathbf{X}$ includes the collection of potential confounders ($\mathbf{X}_c$) and one of the selected biomarkers ($X_b$)(i.e., serum albumin, creatinine, calcium, phosphate, and blood urea nitrogen). To this end, longitudinal biomarker measurements, $X_b$, were incorporated into the Cox PH model in two ways: (i) using only the baseline values at peritoneal dialysis (PD) initiation (M1), and (ii) using time-averaged values over the follow-up period (M2).

In addition to the Cox PH models, M1 and M2, the effects of longitudinally measured biomarkers on survival were examined using a time-dependent Cox PH model (M3), as

**Table 3. Specifications of fitted models.**

| Specifications | Models | | | | |
| --- | --- | --- | --- | --- | --- |
| | M1 | M2 | M3 | M4 | M5 |
| Response | | | | | |
| Survival submodel | Univariate | Univariate | Univariate | Univariate | Univariate |
| Longitudinal submodel | - | - | - | Univariate | Multivariate |
| Time-varying predictors | No | No | Yes | Yes | Yes |
| Submodels | Cox PH | Cox PH | Cox PH | LME & Cox PH | LME & Cox PH |
| Ajdustment on time-varying biomarkers | No | No | No | Yes (via LME) | Yes (via LME) |
| Dynamic predictions | No | No | No | Yes | Yes |
| Number of explicit models fitted | 5 | 5 | 5 | 5 | 1 |

defined in Eq (2), accounting for their time-varying effects.

$$h(t \mid \mathbf{X}_c, X_b(t)) = h_0(t) \cdot \exp\left(\mathbf{X}_c\beta_c + X_b(t)\beta_b\right) \qquad (2)$$

Here, $X_b(t)$ represents one of the longitudinal biomarkers, incorporated as time-varying covariate into model.

**Joint Models (Univariate and Multivariate, i.e., M4 and M5).** The relationship between longitudinal biomarker trajectories and mortality in peritoneal dialysis (PD) patients was evaluated using univariate (M4) and multivariate (M5) joint models. Each joint model integrates two submodels: (i) a linear mixed-effects model (LME) to model the longitudinal biomarker trajectory, and (ii) a Cox PH model to assess mortality risk. This approach estimates the effect of time-varying biomarkers on survival while adjusting for potential confounders and accounting for the correlation between longitudinal and survival submodels [51]. The analyses were conducted in R using the `JMbayes2` package (version 0.5-0) [27].

The univariate joint model (M4) is defined by the survival submodel in Eq (3) and the LME submodel in Eq (4). Here, the survival submodel resembles the time-dependent covariates as in Eq (2), but incorporates error-free biomarker trajectories ($m(t)$) adjusted via the LME submodel.

$$h(t \mid \mathbf{X}_c, m(t)) = h_0(t) \cdot \exp\left(\mathbf{X}_c\beta_c + \alpha m(t)\right) \qquad (3)$$

$$X_b(t) = m(t) + \varepsilon_i = \mathbf{W}\gamma + \mathbf{Z}\omega + \epsilon \qquad (4)$$

In joint models, the time-varying effects of biomarkers are modeled in the LME submodel using fixed (**W**) and random (**Z**) effects, allowing flexible selection of confounders tailored to each longitudinal biomarker. The LME submodel incorporates random intercepts and slopes (via time) for each biomarker, with all other confounders included as fixed effects. This flexibility enables different confounder sets for each biomarker's longitudinal submodel in M4, enhancing model adaptability. For M4, separate joint models were fitted for each biomarker.

The multivariate joint model (M5) extends M4 by simultaneously modeling all biomarkers as a multivariate response ($\mathbf{X}_b(t)$) in Eq (4), with their time-varying effects incorporated as a multivariate component ($m(t)\alpha$) in the survival submodel. Unlike M4, M5 evaluates the combined effect of all biomarkers in a single model.

Parameter estimation in joint models employed a Bayesian approach, optimizing parameters while accounting for covariance structures between the longitudinal and survival submodels. Detailed mathematical backgrounds, including parameter estimation procedures, variable selection strategies, statistical inferences and asymptotic properties of joint models M4 and M5, can be found in the related literature [52–55].

## B. Parameter estimates of fitted models

This appendix presents tables summarizing the results of the fitted models (M1–M5). Table 4 provides the full names of abbreviations used in the model outputs. Tables 5 to 9 report the Cox PH results for models M1 to M3, presented separately for each longitudinal biomarker (serum albumin, blood urea nitrogen, creatinine, calcium, and phosphorus). Univariate joint modeling results for M4 are detailed in Tables 10 to 14, corresponding to each biomarker. Multivariate joint modeling results for M5 are summarized in Table 15.

**Table 4. List of abbreviations used in fitted model formula.**

| Abbreviation | Full name | Unit |
|---|---|---|
| BMI | Body mass index | *kg/m²* |
| BUN | Blood urea nitrogen | *mg/dL* |
| HD | Hemodialysis history | *Yes* or *No* |
| TC | Peritoneal membrane transport characteristics | *Low* or *High* |
| TCRD | Total number of comorbid and renal diseases | *count* |

**Table 5. Parameter estimates of Cox PH models, i.e., M1, M2, and M3 – Serum albumin trajectories.**

| | Models | | | | | |
|---|---|---|---|---|---|---|
| | **Baseline (M1)** | | **Averaged (M2)** | | **Time-dependent (M3)** | |
| **Parameters** | **Estimate** | **HR (95% CI)** | **Estimate** | **HR (95% CI)** | **Estimate** | **HR (95% CI)** |
| Age | 0.037 | 1.037 (1.017, 1.059) | 0.027 | 1.027 (1.008, 1.047) | 0.024 | 1.024 (1.005, 1.043) |
| BMI | 0.047 | 1.048 (0.986, 1.115) | 0.072 | 1.075 (1.012, 1.141) | 0.080 | 1.084 (1.022, 1.149) |
| HD history (Yes) | 0.695 | 2.004 (1.076, 3.735) | 0.914 | 2.495 (1.440, 4.324) | 0.946 | 2.576 (1.492, 4.447) |
| TCRD | 0.487 | 1.627 (1.261, 2.101) | 0.420 | 1.522 (1.175, 1.972) | 0.399 | 1.491 (1.149, 1.934) |
| Peritonit rate | 0.737 | 2.088 (1.493, 2.922) | 0.622 | 1.864 (1.324, 2.623) | 0.576 | 1.778 (1.267, 2.496) |
| Serum albumin | -0.430 | 1.537[†] (1.089, 2.169) | -0.699 | 2.012[†] (1.122, 3.610) | -0.771 | 2.162[†] (1.490, 3.135) |

**HR**: Hazard ratio; **CI**: Confidence interval.
[†]Risk estimates were given for 1-unit decrease in measured variable if found statistically significant.

**Results for Cox PH models, i.e., M1, M2, and M3.**

**Table 6. Parameter estimates of Cox PH models, i.e., M1, M2, and M3 – Creatinine trajectories.**

| | Models | | | | | |
|---|---|---|---|---|---|---|
| | **M1** | | **M2** | | **M3** | |
| **Parameters** | **Estimate** | **HR (95% CI)** | **Estimate** | **HR (95% CI)** | **Estimate** | **HR (95% CI)** |
| Age | 0.031 | 1.032 (1.011, 1.053) | 0.013 | 1.013 (0.993, 1.034) | 0.018 | 1.018 (0.998, 1.038) |
| BMI | 0.054 | 1.056 (0.993, 1.122) | 0.093 | 1.097 (1.033, 1.165) | 0.085 | 1.089 (1.026, 1.155) |
| HD history (Yes) | 1.055 | 2.873 (1.519, 5.433) | 1.198 | 3.312 (1.885, 5.820) | 1.088 | 2.969 (1.711, 5.151) |
| TCRD | 0.484 | 1.622 (1.249, 2.106) | 0.311 | 1.366 (1.048, 1.778) | 0.365 | 1.441 (1.110, 1.869) |
| Peritonit rate | 0.712 | 2.038 (1.454, 2.858) | 0.656 | 1.927 (1.377, 2.699) | 0.653 | 1.923 (1.378, 2.684) |
| Creatinine | -0.063 | 0.939 (0.853, 1.034) | -0.229 | 1.257[†] (1.126, 1.406) | -0.152 | 1.164[†] (1.064, 1.274) |

**HR**: Hazard ratio; **CI**: Confidence interval.
[†]Risk estimates were given for 1-unit decrease in measured variable if found statistically significant.

**Table 7. Parameter estimates of Cox PH models, i.e., M1, M2, and M3 – BUN trajectories.**

| | Models | | | | | |
|---|---|---|---|---|---|---|
| | **M1** | | **M2** | | **M3** | |
| **Parameters** | **Estimate** | **HR (95% CI)** | **Estimate** | **HR (95% CI)** | **Estimate** | **HR (95% CI)** |
| Age | 0.034 | 1.035 (1.014, 1.056) | 0.026 | 1.027 (1.007, 1.046) | 0.027 | 1.027 (1.008, 1.047) |
| BMI | 0.051 | 1.052 (0.989, 1.119) | 0.077 | 1.080 (1.016, 1.149) | 0.075 | 1.078 (1.016, 1.144) |
| HD history (Yes) | 0.905 | 2.473 (1.363, 4.485) | 1.018 | 2.768 (1.597, 4.796) | 0.993 | 2.700 (1.599, 4.677) |
| TCRD | 0.517 | 1.677 (1.299, 2.165) | 0.445 | 1.561 (1.209, 2.016) | 0.454 | 1.574 (1.222, 2.028) |
| Peritonit rate | 0.709 | 2.031 (1.453, 2.841) | 0.653 | 1.921 (1.372, 2.689) | 0.655 | 1.925 (1.378, 2.687) |
| BUN | -0.0005 | 0.999 (0.987, 1.012) | -0.014 | 0.986 (0.966, 1.007) | -0.010 | 0.990 (0.977, 1.003) |

**HR**: Hazard ratio; **CI**: Confidence interval.

**Table 8. Parameter estimates of Cox PH models, i.e., M1, M2, and M3 – Calcium trajectories.**

| Parameters | Models | | | | | |
|---|---|---|---|---|---|---|
| | M1 | | M2 | | M3 | |
| | Estimate | HR (95% CI) | Estimate | HR (95% CI) | Estimate | HR (95% CI) |
| Age | 0.036 | 1.037 (1.016, 1.059) | 0.029 | 1.029 (1.009, 1.050) | 0.028 | 1.029 (1.009, 1.049) |
| BMI | 0.047 | 1.048 (0.986, 1.114) | 0.065 | 1.067 (1.006, 1.131) | 0.066 | 1.068 (1.007, 1.132) |
| HD history (Yes) | 0.916 | 2.499 (1.381, 4.523) | 1.009 | 2.743 (1.589, 4.734) | 1.009 | 2.743 (1.590, 4.734) |
| TCRD | 0.512 | 1.668 (1.292, 2.154) | 0.459 | 1.584 (1.225, 2.048) | 0.465 | 1.592 (1.234, 2.053) |
| Peritonit rate | 0.709 | 2.031 (1.451, 2.843) | 0.667 | 1.948 (1.395, 2.720) | 0.664 | 1.943 (1.393, 2.710) |
| Calcium | -0.122 | 0.885 (0.715, 1.093) | -0.117 | 0.889 (0.617, 1.282) | -0.056 | 0.945 (0.747, 1.196) |

**HR**: Hazard ratio; **CI**: Confidence interval.

**Table 9. Parameter estimates of Cox PH models, i.e., M1, M2, and M3 – Phosphorus trajectories.**

| Parameters | Models | | | | | |
|---|---|---|---|---|---|---|
| | M1 | | M2 | | M3 | |
| | Estimate | HR (95% CI) | Estimate | HR (95% CI) | Estimate | HR (95% CI) |
| Age | 0.034 | 1.035 (1.014, 1.056) | 0.029 | 1.029 (1.009, 1.050) | 0.025 | 1.025 (1.005, 1.045) |
| BMI | 0.052 | 1.053 (0.990, 1.120) | 0.065 | 1.067 (1.006, 1.131) | 0.081 | 1.084 (0.020, 1.152) |
| HD history (Yes) | 0.919 | 2.507 (1.356, 4.635) | 1.009 | 2.743 (1.589, 4.734) | 1.110 | 3.035 (1.730, 5.326) |
| TCRD | 0.517 | 1.677 (1.300, 2.164) | 0.459 | 1.584 (1.225, 2.048) | 0.466 | 1.593 (1.238, 2.049) |
| Peritonit rate | 0.716 | 2.047 (1.451, 2.886) | 0.667 | 1.948 (1.395, 2.720) | 0.709 | 2.033 (1.446, 2.859) |
| Phosphorus | -0.013 | 0.987 (0.861, 1.131) | -0.117 | 0.889 (0.617, 1.282) | -0.159 | 0.853 (0.712, 1.020) |

**HR**: Hazard ratio; **CI**: Confidence interval.

**Table 10. Parameter estimates of univariate joint model (M4) – Serum albumin trajectories.**

*Longitudinal part (Linear Mixed Effects, LME)*

| Parameters | Estimate (95% CI) | p-value |
|---|---|---|
| Age | -0.006 (-0.010, -0.003) | <0.001 |
| TC | -0.358 (-0.445, -0.269) | <0.001 |
| Time | -0.007 (-0.009, -0.006) | <0.001 |

*Survival part (Cox PH)*

| Parameters | Estimate | HR (95% CI) |
|---|---|---|
| Age | 0.024 | 1.024 (1.002, 1.048) |
| BMI | 0.084 | 1.088 (1.020, 1.158) |
| HD history (Yes) | 0.989 | 2.689 (1.496, 4.621) |
| TCRD | 0.406 | 1.501 (1.159, 1.950) |
| Peritonit rate | 0.568 | 1.765 (1.244, 2.406) |
| Serum albumin | -1.066 | 2.904[†] (1.657, 5.132) |

**HR**: Hazard ratio; **CI**: Confidence interval.
[†]Risk estimates were given for 1-unit decrease in measured variable if found statistically significant.

**Table 11. Parameter estimates of univariate joint model (M4) – Creatinine trajectories.**

*Longitudinal part (Linear Mixed Effects, LME)*

| Parameters | Estimate (95% CI) | p-value |
|---|---|---|
| Age | -0.077 (-0.096, -0.059) | <0.001 |
| TC | -1.131 (-1.633, -0.637) | <0.001 |
| TCRD | -0.449 (-0.742, -0.159) | 0.003 |
| HD history (Yes) | 1.285 (0.489, 2.045) | 0.002 |
| Time | 0.052 (0.043, 0.062) | <0.001 |

*Survival part (Cox PH)*

| Parameters | Estimate | HR (95% CI) |
|---|---|---|
| Age | 0.020 | 1.020 (0.994, 1.047) |
| BMI | 0.087 | 1.091 (1.022, 1.160) |
| HD history (Yes) | 1.127 | 3.088 (1.745, 5.315) |
| TCRD | 0.388 | 1.474 (1.137, 1.914) |
| Peritonit rate | 0.638 | 1.893 (1.366, 2.593) |
| Creatinine | -0.145 | 1.156[†] (1.040, 1.292) |

**HR**: Hazard ratio; **CI**: Confidence interval.
[†]Risk estimates were given for 1-unit decrease in measured variable if found statistically significant.

**Table 12. Parameter estimates of univariate joint model (M4) – Phosphorus trajectories.**

*Longitudinal part (Linear Mixed Effects, LME)*

| Parameters | Estimate (95% CI) | p-value |
|---|---|---|
| BMI | 0.057 (0.026, 0.088) | <0.001 |
| Age | -0.022 (-0.030, -0.013) | <0.001 |
| TC | -0.459 (-0.696, -0.222) | <0.001 |
| Time | 0.003 (-0.002, 0.007) | 0.242 |

*Survival part (Cox PH)*

| Parameters | Estimate | HR (95% CI) |
|---|---|---|
| Age | 0.024 | 1.024 (0.999, 1.049) |
| BMI | 0.088 | 1.092 (1.023, 1.162) |
| HD history (Yes) | 1.122 | 3.071 (1.806, 5.228) |
| TCRD | 0.473 | 1.605 (1.222, 2.056) |
| Peritonit rate | 0.689 | 1.993 (1.440, 2.737) |
| Phosphorus | -0.219 | 1.245† (1.021, 1.540) |

**HR**: Hazard ratio; **CI**: Confidence interval.
†Risk estimates were given for 1-unit decrease in measured variable if found statistically significant.

**Table 13. Parameter estimates of univariate joint model (M4) – BUN trajectories.**

*Longitudinal part (Linear Mixed Effects, LME)*

| Parameters | Estimate (95% CI) | p-value |
|---|---|---|
| BMI | 0.948 (0.631, 1.260) | <0.001 |
| Age | -0.162 (-0.255, -0.070) | 0.001 |
| TC | -2.408 (-4.808, -0.030) | 0.046 |
| TCRD | -1.635 (-2.973, -0.259) | 0.020 |
| Time | 0.036 (-0.020, 0.089) | 0.192 |

*Survival part (Cox PH)*

| Parameters | Estimate | HR (95% CI) |
|---|---|---|
| Age | 0.026 | 1.026 (1.002, 1.052) |
| BMI | 0.083 | 1.086 (1.188, 1.158) |
| HD history (Yes) | 1.012 | 2.750 (1.582, 4.721) |
| TCRD | 0.456 | 1.577 (1.207, 2.073) |
| Peritonit rate | 0.630 | 1.878 (1.339, 2.555) |
| BUN | -0.017 | 0.983 (0.959, 1.000) |

**HR**: Hazard ratio; **CI**: Confidence interval

**Table 14. Parameter estimates of univariate joint model (M4) – Calcium trajectories.**

| *Longitudinal part (Linear Mixed Effects, LME)* | | |
|---|---|---|
| **Parameters** | **Estimate (95% CI)** | **p-value** |
| TC | -0.224 (-0.352, -0.096) | <0.001 |
| Time | 0.003 (0.001, 0.005) | 0.009 |
| *Survival part (Cox PH)* | | |
| **Parameters** | **Estimate** | **HR (95% CI)** |
| Age | 0.03 | 1.030 (1.006, 1.054) |
| BMI | 0.066 | 1.068 (1.000, 1.137) |
| HD history (Yes) | 1.036 | 2.817 (1.616, 4.720) |
| TCRD | 0.472 | 1.602 (1.237, 2.088) |
| Peritonit rate | 0.654 | 1.923 (1.453, 2.598) |
| Calcium | -0.094 | 0.910 (0.726, 1.664) |

**HR**: Hazard ratio; **CI**: Confidence interval.

The multivariate joint model (M5) was used to evaluate the combined effects of all longitudinal biomarkers on mortality. For each biomarker, a LME submodel was fitted, incorporating time as a random slope and confounders as fixed effects. These LME submodels were integrated into a single multivariate joint model, combining them with a Cox PH submodel to account for mortality, while modeling the covariance structure of random intercepts and slopes across biomarkers. The LME submodels for the longitudinal biomarkers are specified as follows:

$$
\begin{aligned}
ALB_{ij(1)} &= \gamma_{0(1)} + \gamma_{1(1)} Age + \gamma_{2(1)} TC + \gamma_{1(1)} t_{ij(1)} \\
&\quad + \omega_{0i(1)} + \omega_{1i(1)} t_{ij(1)} + \epsilon_{ij(1)} \\
BUN_{ij(2)} &= \gamma_{0(2)} + \gamma_{1(2)} Age + \gamma_{2(2)} BMI + \gamma_{3(2)} TCRD + \gamma_{4(2)} TC + \gamma_{5(2)} t_{ij(2)} \\
&\quad + \omega_{0i(2)} + \omega_{1i(2)} t_{ij(2)} + \epsilon_{ij(2)} \\
CR_{ij(3)} &= \gamma_{0(3)} + \gamma_{1(3)} Age + \gamma_{2(3)} HD + \gamma_{3(3)} TCRD + \gamma_{4(3)} TC + \gamma_{5(3)} t_{ij(3)} \\
&\quad + \omega_{0i(3)} + \omega_{1i(3)} t_{ij(3)} + \epsilon_{ij(3)} \\
CA_{ij(4)} &= \gamma_{0(4)} + \gamma_{1(4)} TC + \gamma_{2(4)} t_{ij(4)} + \omega_{0i(4)} + \omega_{1i(4)} t_{ij(4)} + \epsilon_{ij(4)} \\
P_{ij(5)} &= \gamma_{0(5)} + \gamma_{1(5)} Age + \gamma_{2(5)} BMI + \gamma_{3(5)} TC + \gamma_{4(5)} t_{ij(5)} \\
&\quad + \omega_{0i(5)} + \omega_{1i(5)} t_{ij(5)} + \epsilon_{ij(5)}
\end{aligned}
\tag{5}
$$

In Eq (5), $i$ denotes individual patients, $j$ denotes time points, and $k$ = 1, 2, 3, 4, 5 corresponds to the $k$-th biomarker. The parameters $\gamma_{.(k)}$ represent fixed effects of predictors (including confounders and time) for the $k$-th biomarker, $\omega_{0i(k)}$ and $\omega_{1i(k)}$ denote random intercepts and slopes for the $i$-th patient, and $\epsilon_{ij(k)}$ is the random error term. Results of the multivariate joint model (M5) are summarized in Table 15.

**Results for Joint Models, i.e., M4 and M5.**

**Table 15. Parameter estimates of multivariate joint model (M5).**

*Longitudinal part (Linear Mixed Effects, LME)*

| Parameters | Estimate (95% CI) | p-value |
|---|---|---|
| *Serum albumin* | | |
| Age | -0.007 (-0.010, -0.004) | <0.001 |
| TC | -0.366 (-0.451, -0.281) | <0.001 |
| Time | -0.008 (-0.010, -0.006) | <0.001 |
| *Blood urea nitrogen (BUN)* | | |
| BMI | 0.713 (0.423, 1.010) | <0.001 |
| Age | -0.142 (-0.236, -0.049) | 0.004 |
| TC | -2.786 (-5.124, -0.397) | 0.022 |
| TCRD | -1.506 (-2.865, -0.151) | 0.029 |
| Time | 0.061 (-0.001, 0.128) | 0.052 |
| *Creatinine* | | |
| Age | -0.078 (-0.096, -0.059) | <0.001 |
| TC | -1.120 (-1.620, -0.630) | <0.001 |
| TCRD | -0.413 (-0.671, -0.150) | 0.003 |
| HD history (Yes) | 0.872 (0.214, 1.542) | 0.011 |
| Time | 0.054 (0.045, 0.064) | <0.001 |
| *Calcium* | | |
| TC | -0.227 (-0.353, -0.100) | <0.001 |
| Time | 0.002 (0.001, 0.004) | 0.201 |
| *Phosphorus* | | |
| BMI | 0.041 (0.015, 0.069) | 0.001 |
| Age | -0.020 (-0.030, -0.011) | <0.001 |
| TC | -0.473 (-0.703, -0.239) | <0.001 |
| Time | 0.003 (-0.002, 0.007) | 0.202 |

*Survival part (Cox PH)*

| Parameters | Estimate | HR (95% CI) |
|---|---|---|
| Age | 0.014 | 1.014 (0.986, 1.041) |
| BMI | 0.112 | 1.119 (1.047, 1.194) |
| HD history (Yes) | 1.032 | 2.806 (1.492, 5.046) |
| TCRD | 0.347 | 1.414 (1.069, 1.857) |
| Peritonit rate | 0.538 | 1.713 (1.197, 2.399) |
| Serum albumin | -1.191 | 3.291[†] (1.639, 7.229) |
| BUN | -0.010 | 0.990 (0.965, 1.015) |
| Creatinine | -0.094 | 0.910 (0.798, 1.036) |
| Calcium | 0.280 | 1.324 (0.799, 2.207) |
| Phosphorus | -0.021 | 0.979 (0.790, 1.291) |

**HR**: Hazard ratio; **CI**: Confidence interval.
[†]Risk estimates were given for 1-unit decrease in measured variable if found statistically significant.

## Supporting information

**S1 File. A ZIP file including the analysis codes.** Analysis code has been included in the supplementary materials. Additionally, all code is freely accessible via GitHub at https://github.com/basolmerve/MultivariateJM_Supplementary.
(ZIP)

## Author contributions

**Conceptualization:** Merve Basol Goksuluk, Dincer Goksuluk, Murat Hayri Sipahioglu.

**Data curation:** Merve Basol Goksuluk, Murat Hayri Sipahioglu.

**Formal analysis:** Merve Basol Goksuluk, Dincer Goksuluk.

**Investigation:** Merve Basol Goksuluk, Murat Hayri Sipahioglu.

**Methodology:** Merve Basol Goksuluk, Dincer Goksuluk.

**Project administration:** Murat Hayri Sipahioglu.

**Resources:** Merve Basol Goksuluk.

**Software:** Merve Basol Goksuluk, Dincer Goksuluk.

**Supervision:** Murat Hayri Sipahioglu.

**Validation:** Merve Basol Goksuluk, Dincer Goksuluk, Murat Hayri Sipahioglu.

**Visualization:** Merve Basol Goksuluk, Dincer Goksuluk.

**Writing – original draft:** Merve Basol Goksuluk, Dincer Goksuluk.

**Writing – review & editing:** Merve Basol Goksuluk, Dincer Goksuluk.

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
