## [Decision Letter · Decision Letter 0]

10 Apr 2025

PONE-D-25-02999Assessing Biomarker Trajectories for Mortality Risk in Peritoneal Dialysis: A Focus on Multivariate Joint ModelingPLOS ONE

Dear Dr. Goksuluk,

Thank you for submitting your manuscript to PLOS ONE. After careful consideration, we feel that it has merit but does not fully meet PLOS ONE’s publication criteria as it currently stands. Therefore, we invite you to submit a revised version of the manuscript that addresses the points raised during the review process.

We look forward to receiving your revised manuscript.

Kind regards,

Ankur Shah

Academic Editor

PLOS ONE

3. In the online submission form, you indicated that [The peritoneal dialysis data used to support the �ndings of this study are available from the corresponding author upon request.].

4.  We are unable to open your Figure file [fig1.eps, dynPred.eps and AUCPlots.eps]. Please kindly revise as necessary and re-upload.

Additional Editor Comments:

The manuscript is Of interest but the reviewers highlight areas for improvement. Review their comments and respond.

Reviewers' comments:

Reviewer's Responses to Questions

**Comments to the Author**

1. Is the manuscript technically sound, and do the data support the conclusions?

Reviewer #1: Partly

Reviewer #2: Partly

2. Has the statistical analysis been performed appropriately and rigorously? 

Reviewer #1: No

Reviewer #2: N/A

3. Have the authors made all data underlying the findings in their manuscript fully available?

Reviewer #1: No

Reviewer #2: No

4. Is the manuscript presented in an intelligible fashion and written in standard English?

Reviewer #1: Yes

Reviewer #2: Yes

5. Review Comments to the Author

Reviewer #1: The authors analyzed mortality risk data with a restrospective cohort of 417 peritoneal dialysis patients followed-up between 1995 and 2016 using both traditional and advanced statistical approaches. They concluded that joint models outperformed other traditional approaches. Also, serum albumin, creatine and phosphorus showed significance as mortality predictors.

1. Inclusion and exclusion criteria. Did the study include any women who pregnant?

2. Two patients were selected and excluded to serve as the test set. Please provide more in detail information on this design and rational.

3. Continuous variables were summarized depending on the data distribution. Please be more specific.

4. Please clarify whether the proportional hazards assumptions were evaluated. What if the assumptions were not satisfied.

5. It’s unclear how M2 (average value) were conduct in a Cox PH model as the values have been averaged.

6. In equation 1, many abbreviations have been used without explanation, except BUN.

7. Why wasn’t Age adjusted as a random effect in CR, CA, and PH analysis?

8. It’s nice to see a clear pattern that joint modeling predicted the survival rate more aligned to the two selected individuals based on Figure 2. However, how do we know whether this is more due to the chance. For example, in Figure 1, the trajectories from two selected individuals looked quite different from two selected individuals.

9. The conclusion on the method comparison based on one data without simulation is less convincing without some rigorous simulation studies.

Reviewer #2: The authors present a retrospective study investigating mortality risk prediction in PD patients through longitudinal biomarker analysis, comparing traditional and advanced statistical approaches. They showed a highest predictive accuracy of multivariate joint modeling.

I suggest the following revisions:

1. Rationale should be strengthened – for example, how about the current progression analysis of PD patients, including prognostic models and predictors? The use of dynamic prediction models in other chronic disease management? And are there any examples of actual benefits brought by the evaluation of biomarkers trajectories in clinical actions?

2. Please explain the reason why patients with survival of fewer than 90 days after starting PD were excluded, and whether any patients with other fatal diseases such as malignant tumor or severe cardiovascular disease were excluded?

3. The Charlson Comorbidity Index should be used in the statistical analyses to describe the overall comorbidity.

4. It is not clear from the methodology section how the authors defined “TCRD” or “number of illnesses”?

5. Please show the precise results of Cox PH regression models for the time-to-event analysis (as mentioned in line152, page 5 and line 153-155 in page7), and it’s not clear how the “TC” were employed into the model as a variable.

6. Please clarify the results of M1-M3 in table2, whether it’s univariate model or multivariate model.

7. It’s confused how the two patients (i.e. test samples) were chosen from the complete dataset? Have they been selected randomly?

8. Please show the study limitations, which is essential for the study generalization.

6. PLOS authors have the option to publish the peer review history of their article (what does this mean?). If published, this will include your full peer review and any attached files.

Reviewer #1: No

Reviewer #2: No

---

## [Author Response · Author response to Decision Letter 1]

2 Jun 2025

Response to Editorial Comments

Response: Thank you for your comment. We tried our best to follow journal’s style in the revised manuscript.

Response: Thank you for your comment. All analysis code used in this study has been provided as supplementary material. In addition, to promote transparency, reproducibility, and ease of reuse, we have made the full set of analysis scripts publicly available via GitHub at the following URL: https://github.com/basolmerve/MultivariateJM_Supplementary.

This information has also been clearly stated in the Data Availability section of the manuscript, in accordance with PLOS ONE’s code sharing policy.

3. In the online submission form, you indicated that [The peritoneal dialysis data used to support the �ndings of this study are available from the corresponding author upon request.].

Response: Thank you for your valuable feedback and for highlighting the data availability policy of PLOS journals. We appreciate the importance of ensuring that research data are accessible to the scientific community. However, we must address specific ethical and legal constraints related to the peritoneal dialysis dataset used in this study.

The dataset includes personal information and clinical data of patients who remain under surveillance, subjecting it to the General Data Protection Regulation (GDPR) and other privacy regulations. Publicly sharing this dataset would compromise patient privacy and violate the protocol approved by our research ethics board.

To reconcile the need for data accessibility with these obligations, we propose an alternative approach. The data can be shared with applicant upon request after removing patient-specific identifiers to ensure anonymity, a process that still requires prior permission from our Institutional Executive Board. Researchers interested in accessing the data may contact the senior author/data holder (Murat Hayri Sipahioglu), and such requests will be forwarded to the Executive Board for approval.

We therefore respectfully request an exemption from the requirement to deposit the data publicly. Instead, we will include a statement in the manuscript specifying that the data are available from the senior author upon reasonable request, contingent upon approval by the Institutional Executive Board. We think this approach upholds the principles of open science while adhering to our ethical and legal responsibilities.

We included below paragraph under “Data Availability” section:

“The peritoneal dialysis data supporting the findings of this study are available from the Department of Internal Medicine, Division of Nephrology, at Erciyes University (via https://tip.erciyes.edu.tr) upon reasonable request and subject to approval by the Institutional Executive Board, due to ethical and legal restrictions related to patient privacy and General Data Protection Regulation (GDPR) requirements. Researchers may also contact the senior author (Dr. Murat Hayri Sipahioglu, https://avesis.erciyes.edu.tr/mhsipahioglu), who can facilitate the data access process by forwarding requests to the appropriate institutional committee.”

We are grateful for your understanding and welcome any further questions or suggestions you may have.

4. We are unable to open your Figure file [fig1.eps, dynPred.eps and AUCPlots.eps]. Please kindly revise as necessary and re-upload.

Response: Thank you for your comment, and we apologize for the technical issues you encountered. We have uploaded the original EPS files to the PACE system and additionally converted the corresponding figures to TIFF format, as required. To ensure compatibility, we have submitted all figures in three formats: EPS, TIFF, and PDF. If there are still any issues accessing the figures, please do not hesitate to let us know, and we will be happy to assist further.

Response to Reviewer’s Comments

Reviewer #1: The authors analyzed mortality risk data with a restrospective cohort of 417 peritoneal dialysis patients followed-up between 1995 and 2016 using both traditional and advanced statistical approaches. They concluded that joint models outperformed other traditional approaches. Also, serum albumin, creatine and phosphorus showed significance as mortality predictors.

We would like to sincerely thank the reviewer for their valuable and constructive comments. Their suggestions have significantly contributed to improving the clarity and overall quality of the manuscript. Each point raised has been carefully addressed to the best of our ability, and detailed explanations of the revisions have been provided in our point-by-point responses. The corresponding changes have been made in the manuscript and are clearly marked using track changes. Line references in our responses refer to the tracked version of the revised manuscript.

1. Inclusion and exclusion criteria. Did the study include any women who pregnant?

Response: We did not include any pregnant woman in the study. This criterion is also included in the Exclusion Criteria under Materials and Methods section as “… (iv) pregnancy; ….”. (line 100)

2. Two patients were selected and excluded to serve as the test set. Please provide more in detail information on this design and rational.

Response: Thank you for your valuable comment requesting further details on the design and rationale for selecting and excluding two test patients. To ensure an unbiased evaluation, two patients—one censored and one who died—were randomly selected from the cohort without considering their biomarker trajectories (e.g., albumin, creatinine, calcium, phosphorus, and blood urea nitrogen). These patients were excluded from the training dataset (n=415) to serve as an independent test set, allowing assessment of model performance on unseen data. Unlike traditional survival models, such as Cox regression, which derive population-based predictions, our multivariate joint modeling approach estimates patient-specific survival probabilities through random effects, capturing individual biomarker dynamics. Consequently, evaluating predictive performance at the patient level, as demonstrated with two representative patients, was sufficient to illustrate the models’ capabilities, and a larger test set was unnecessary for this purpose. This design leverages the strength of dynamic, personalized predictions, aligning with our study’s objectives. We have included a concise explanation of this approach in the “Study Design and Participants” section (lines 103 – 109).

3. Continuous variables were summarized depending on the data distribution. Please be more specific.

Response: We selected appropriate summary statistics depending on the normality of continuous variables. To clarify this, we added/modified below text to the “Statistical Analysis” section (lines 142 – 145).

“The normality of continuous variables was assessed using the Shapiro-Wilk test and graphical methods such as histograms, Q-Q plots, and box plots. Normally distributed variables were summarized using means and standard deviations while nonnormally distributed variables were summarized using medians and minimum and maximum values.”

4. Please clarify whether the proportional hazards assumptions were evaluated. What if the assumptions were not satisfied.

Response: Thank you for highlighting this important aspect of survival analysis. We evaluated the proportional hazards (PH) assumption for all fitted survival models using standard diagnostic tests in R programming language (i.e., “cox.zph” function under survival package). All survival models included a consistent set of covariates: age, body mass index (BMI), history of hemodialysis (preHD), total number of renal and comorbid diseases (TCRD), peritoneal membrane transport characteristic (TC), and one of the selected biomarkers—albumin, creatinine, blood urea nitrogen (BUN), phosphorus, or calcium. The results indicated that the PH assumption was not violated globally across the models. However, we observed that the TC covariate exhibited a local violation of the PH assumption in some models. Despite this, it did not compromise the overall model validity in global tests. Given the clinical relevance of TC in the progression of PD outcomes, we retained it in the survival models. Moreover, we are aware that conducting multiple tests for the PH assumption—one for each covariate in each fitted model—can increase the risk of Type I error. For this reason, we based our modeling decisions on the overall pattern of test results rather than isolated violations. In summary, the PH assumption was generally satisfied across the survival models, and no violations were observed that would compromise the validity or interpretability of the joint models used in downstream analyses.

We did not include the detailed test results in the manuscript, as presenting them in a single table would have made the content overly long and difficult to interpret. However, we summarized this information in the Results section (lines 240 – 245).

5. It’s unclear how M2 (average value) were conduct in a Cox PH model as the values have been averaged.

Response: Thank you for your comment. The M2 model is structurally similar to the Cox PH model that incorporates baseline biomarker values (i.e., M1); however, instead of using baseline values, it includes the arithmetic mean of the repeatedly measured biomarkers across the follow-up period. While M2 does not account for the time-dependent trajectory of biomarkers, it captures their average effect over time, offering a simpler alternative for prediction modeling.

To implement M2, we calculated the average value of each longitudinal biomarker for each patient and included these as covariates in the Cox PH model. This approach, although limited in capturing dynamic changes, is commonly used for its interpretability and simplicity. To improve clarity, we have provided detailed information regarding the mathematical formulation and model specifications in the Appendix section of the manuscript (Appendix A and Table 3).

6. In equation 1, many abbreviations have been used without explanation, except BUN.

Response: Thank you for your observation. All abbreviations used in the equations are introduced in the main text upon their first appearance. To maintain readability and avoid interrupting the narrative flow, we have provided the mathematical background and detailed model specifications in the Appendix A section. Additionally, a comprehensive list of abbreviations used in the equations is included as a table in Appendix B, where each term is defined in full for reference.

7. Why wasn’t Age adjusted as a random effect in CR, CA, and PH analysis?

Response: Thank you for drawing attention to this issue. There was a typographical error in the model formulations. The Age variable was not included as a random effect in any of the models. However, we noticed that Age was mistakenly listed alongside the random intercept terms in the ALB and BUN models. We have corrected the model equations accordingly (equation 5 between lines 788 and 789).

To improve clarity and avoid overcomplicating the main text, we have provided all theoretical details and complete model specifications in Appendix A and Appendix B of the manuscript.

8. It’s nice to see a clear pattern that joint modeling predicted the survival rate more aligned to the two selected individuals based on Figure 2. However, how do we know whether this is more due to the chance. For example, in Figure 1, the trajectories from two selected individuals looked quite different from two selected individuals.

Response: Thank you for the insightful comment regarding the performance of our multivariate joint modeling approach in predicting survival rates for the two selected individuals in Figure 2. We acknowledge the concern about whether the observed patterns are due to chance, particularly given the differing trajectories in Figure 1.

The two test patients were randomly selected without prior consideration of their biomarker trajectories (e.g., albumin, creatinine), ensuring an unbiased evaluation. Our dataset includes a small minority of outlier patients whose biomarker trends deviate from clinical expectations, such as those with improving biomarkers who died or deteriorating biomarkers who survived. For such outliers, our model predictably assigns higher survival probabilities to patients with worsening trajectories and higher probabilities to those with improving trends, consistent with what the models have learnt from complete data. This does not indicate poor model performance but reflects the expected limitations in handling rare outliers, which we confirmed some are prevalent in our dataset.

We did not specifically focus on these outliers during model evaluation but ensured their inclusion during training to evaluate the model’s robustness to real-world variability. While selecting a different pair of test patients could yield varying predictions, potentially with weaker performance for specific cases. However, we think that this would not generalize to the entire population, as our model demonstrates strong predictive performance at patients’ level. The distinct trajectories of individual patients underscore the need for personalized dynamic predictions, as our random-effects multivariate joint model provides, rather than relying on population-based models. This highlights the strength of our approach in capturing patient-specific biomarker dynamics for tailored risk assessment.

9. The conclusion on the method comparison based on one data without simulation is less convincing without some rigorous simulation studies.

Response: Thank you for your valuable comment regarding the need for rigorous simulation studies to strengthen our method comparison. Initially, we considered conducting a comprehensive simulation study to enhance the generalizability of our findings. However, the models compared in our study—each with distinct distributional assumptions, particularly in formulating the baseline hazard—posed significant challenges. Specifically, the data generation process would inherently favor the model whose assumptions align with the simulated data’s underlying properties, potentially biasing the comparison. Moreover, the models involve numerous parameters, leading to risks of overparameterization and computational intractability. These complexities, combined with the time-intensive nature of designing a robust simulation framework, prevented us from including such a study. We acknowledge this as a limitation and have added it to the manuscript’s discussion, noting that future research sh

---

## [Decision Letter · Decision Letter 1]

2 Jul 2025

Assessing Biomarker Trajectories for Mortality Risk in Peritoneal Dialysis: A Focus on Multivariate Joint Modeling

PONE-D-25-02999R1

Dear Dr. Goksuluk,

We’re pleased to inform you that your manuscript has been judged scientifically suitable for publication and will be formally accepted for publication once it meets all outstanding technical requirements.

Kind regards,

Ankur Shah

Academic Editor

PLOS ONE

Additional Editor Comments (optional):

the authors were responsive to the reviewers comments and the manuscript is suitable for publication

Reviewers' comments:

Reviewer's Responses to Questions

**Comments to the Author**

1. If the authors have adequately addressed your comments raised in a previous round of review and you feel that this manuscript is now acceptable for publication, you may indicate that here to bypass the “Comments to the Author” section, enter your conflict of interest statement in the “Confidential to Editor” section, and submit your "Accept" recommendation.

Reviewer #1: All comments have been addressed

2. Is the manuscript technically sound, and do the data support the conclusions?

Reviewer #1: (No Response)

3. Has the statistical analysis been performed appropriately and rigorously? 

Reviewer #1: (No Response)

4. Have the authors made all data underlying the findings in their manuscript fully available?

Reviewer #1: (No Response)

5. Is the manuscript presented in an intelligible fashion and written in standard English?

Reviewer #1: (No Response)

6. Review Comments to the Author

Reviewer #1: (No Response)

7. PLOS authors have the option to publish the peer review history of their article (what does this mean?). If published, this will include your full peer review and any attached files.

Reviewer #1: No

---

## [Editor Report · Acceptance letter]

PONE-D-25-02999R1

PLOS ONE

Dear Dr. Goksuluk,

I'm pleased to inform you that your manuscript has been deemed suitable for publication in PLOS ONE. Congratulations! Your manuscript is now being handed over to our production team.

Kind regards,

on behalf of

Dr. Ankur Shah

Academic Editor

PLOS ONE